# Long Non-Coding RNA-Based Functional Prediction Reveals Novel Targets in Notch-Upregulated Ovarian Cancer

**DOI:** 10.3390/cancers14061557

**Published:** 2022-03-18

**Authors:** Seonhyang Jeong, Sunmi Park, Young Suk Jo, Moon Jung Choi, Gibbeum Lee, Seul Gi Lee, Min Chul Choi, Hyun Park, Won Duk Joo, Sang Geun Jung, Jandee Lee

**Affiliations:** 1Department of Internal Medicine, Yonsei University College of Medicine, 50-1 Yonsei-ro, Seodaemun-gu, Seoul 03722, Korea; bambi_89@yuhs.ac (S.J.); sunmip@yuhs.ac (S.P.); joys@yuhs.ac (Y.S.J.); 2Department of Surgery, Open NBI Convergence Technology Research Laboratory, Severance Hospital, Yonsei Cancer Center, Yonsei University College of Medicine, 50-1 Yonsei-ro, Seodaemun-gu, Seoul 03722, Korea; moon05@ibs.re.kr (M.J.C.); gblee@yuhs.ac (G.L.); 3Department of Surgery, Eulji University School of Medicine, 95 Dunsanseo-ro, Seo-gu, Daejeon 35233, Korea; seulgi@eulji.ac.kr; 4Department of Gynecological Oncology, Comprehensive Gynecologic Cancer Center, CHA University, 59 Yatap-ro, Bundang-gu, Seongnam 13496, Korea; oursk79@cha.ac.kr (M.C.C.); p06162006@cha.ac.kr (H.P.); wdjoo@cha.ac.kr (W.D.J.)

**Keywords:** long non-coding RNA, Notch, ovarian cancer, DNA repair, spermatogenesis

## Abstract

**Simple Summary:**

The study contributes to our understanding of the role of lncRNAs in the regulation of Notch signaling and their target genes in ovarian cancer. These lncRNAs were identified with in silico analyses using The Cancer Genome Atlas and the results were validated using our transcriptome data from the in vitro NOTCH1/3 silencing experiments and the qRT-PCR analysis, using tissue samples from patients with ovarian tumors. Finally, to improve our understanding of the complexity of Notch signaling, we identified master transcriptional factors which might orchestrate the upregulation of Notch-related lncRNAs and coding-genes in ovarian cancer.

**Abstract:**

Notch signaling is a druggable target in high-grade serous ovarian cancers; however, its complexity is not clearly understood. Recent revelations of the biological roles of lncRNAs have led to an increased interest in the oncogenic action of lncRNAs in various cancers. In this study, we performed in silico analyses using The Cancer Genome Atlas data to discover novel Notch-related lncRNAs and validated our transcriptome data via NOTCH1/3 silencing in serous ovarian cancer cells. The expression of novel Notch-related lncRNAs was down-regulated by a Notch inhibitor and was upregulated in high-grade serous ovarian cancers, compared to benign or borderline ovarian tumors. Functionally, Notch-related lncRNAs were tightly linked to Notch-related changes in diverse gene expressions. Notably, genes related to DNA repair and spermatogenesis showed specific correlations with Notch-related lncRNAs. Master transcription factors, including EGR1, CTCF, GABPα, and E2F4 might orchestrate the upregulation of Notch-related lncRNAs, along with the associated genes. The discovery of Notch-related lncRNAs significantly contributes to our understanding of the complex crosstalk of Notch signaling with other oncogenic pathways at the transcriptional level.

## 1. Introduction

Ovarian cancer is one of the common types of malignancies in women and is often fatal due to the unavailability of precise early-detection tests, as well as its aggressive features, such as an early progression to metastatic disease [1,2,3,4,5]. More than 60% of patients first diagnosed with ovarian cancer have stage-III or stage-IV disease, indicating that it has spread into the abdominal cavity and other peritoneal structures [6,7,8]. Histologic subtypes of ovarian cancer can also affect clinical outcomes, along with the stage. In this regard, a high-grade serous carcinoma (HGSC) is considered, due to structural and ultrastructural evidence, as the most aggressive subtype of ovarian cancer, with a debated origin, a frequent resistance to chemotherapy, and a poor prognosis [9,10,11,12]. To overcome these diagnostic and therapeutic limitations, clinicians and researchers have investigated the biological and clinical features of ovarian cancer, especially HGSC [10,13].

About 98% of the human genome is composed of protein non-coding regions, being transcribed into non-coding RNAs [14]. Among the non-coding RNAs, long non-coding RNAs (lncRNAs) are defined as transcripts longer than 200 nucleotides. According to the Functional Annotation of the Mouse/Mammalian Genomes (FANTOM), 27,919 lncRNAs have been identified in humans [15,16]. Moreover, recent transcriptomic analyses by next generation sequencing have indicated that the number of lncRNAs in mammals could be in the order of tens of thousands [17]. Due to the rapid development in next generation sequencing methods, various lncRNAs, such as ANRIL, CCAT1, FAL1, H19, MALAT1, and HOTAIR, have been discovered in ovarian cancer [18]. HOTAIR has been recently discovered to enhance the invasion and migration of ovarian cancer progression by regulating EZH2, as well as regulating the expression of miR-193a and ODK2, via introducing H3K27me3 [19]. However, despite accumulating evidence suggesting that the majority of lncRNAs are biologically relevant, the insight into their functions in cell biology is still limited.

Over the past two decades, the major components of Notch signaling pathways, including Notch receptors 1 and 3 (NOTCH1 and NOTCH3); their ligands, such as jagged canonical Notch ligand 1 (JAG1) and delta like canonical Notch ligand 4 (DLL4); and downstream proteins, such as the Hes family bHLH transcription factor 1 (HES1) and DLG associated protein 5 (DLGAP5), have been intensively studied in ovarian cancers [20,21,22]. The Notch signaling pathway has important roles in angiogenesis and chemo-resistance. Additionally, Notch signaling inhibitors, such as γ-secretase inhibitors (GSIs) and DLL4 inhibitors, as well as Notch antibodies, have been examined in preclinical and clinical settings [23,24]. However, more work is needed to understand the diverse cell signaling pathways and cell biological processes that are affected by Notch activation in order to develop more specific and effective means of inhibiting Notch pathway components [25]. In this regard, Notch-related lncRNAs have been described in other cancers [26].

In the present study, we identified potential Notch-related lncRNA clusters in ovarian cancer (OVCA) using data from The Cancer Genome Atlas (TCGA) and our transcriptome analysis [27,28,29]. We also compared the cell biological functions of Notch-related mRNAs proposed by a gene set enrichment analysis (GSEA) and that of Notch-related lncRNAs identified by the in silico prediction of lncRNA function (FuncPred) [17,30]. In addition, to understand the regulatory mechanism of Notch-related lncRNAs, we examined master transcriptional factors (TFs) using the Encyclopedia of DNA Elements (ENCODE) [31]. This study provides a fresh insight into how Notch-related lncRNAs could cooperate with Notch-related gene transcripts through novel signaling mechanisms.

## 2. Materials and Methods

### 2.1. Public Databases

Two representative public datasets were utilized in this study. First, the mRNA expression dataset from TCGA (https://portal.gdc.cancer.gov/ (access on 1 November 2021)) OVCA RNA-seq database (http://cancergenome.nih.gov/ (access on 1 November 2021), version 1.0.6) was collected and analyzed using GSEA, with hallmark gene sets of the Molecular Signatures Database (MSigDB), gene sets derived from the Kyoto Encyclopedia of Genes and Genomes (KEGG), and gene sets related to TF targets based on sharing upstream cis-regulatory motifs as potential TF binding sites [30,32]. For a large number of gene sets, nominal *p*-values were considered to indicate a significant enrichment of a set if its normalized enrichment scores (NES) had a False Discovery Rate (FDR) q-value below 0.25. Second, to identify Notch-related lncRNA, its expression dataset was collected from The Atlas of Non-coding RNAs in Cancer (TANRIC) and was analyzed for its functional aspects using FuncPred (http://www.funcpred.com/ (access on 17 August 2021) [17,29]. The information on gene regulation, including the binding site and TFs of Notch-related lncRNA, was investigated with ENCODE data using the Ensembl Genome Browser. The ENCODE analysis, based on chromatin immunoprecipitation sequencing data, was used to identify the regulatory elements for each gene; scores were presented in the ENCODE table. The higher the score, the stronger the binding of the relevant motif, and the greater likelihood of it being identified in all cell lines [31]. The overall schematic flow of our research strategy is presented in Appendix A.

### 2.2. Patients and Specimens

Human tissue samples were obtained from 100 female patients (mean age = 54.4 ± 9.1) following a unilateral or bilateral salpingo-oophorectomy for benign or borderline tumors (*n* = 16), clear cell carcinoma (*n* = 5), or HGSC (*n* = 79) between October 2018 and May 2020 at Bundang CHA Medical Center (CHA University, Gyeonggi-do, Korea). All samples were frozen in liquid nitrogen and stored at −80 °C until use. This study was approved by the institutional review board of Bundang CHA Medical Center (CHAMC 2018-10-008, 10 October 2018) and was conducted in accordance with the recommendations of the institutional review board. Informed consent was obtained from all individual participants included in this study.

### 2.3. RNA Isolation and Real-Time PCR

Total RNA was isolated from frozen tissues using a Trizol reagent (Invitrogen, Carlsbad, CA, USA), and the RNA quality was assessed using a 2100 Bioanalyzer System (Agilent Technologies, Santa Clara, CA, USA). Complementary DNA (cDNA) was prepared from the total RNA using the SuperScript™ III First-Strand Synthesis System (Invitrogen, 18080051). Quantitative real-time PCR (qRT-PCR) was performed using a QuantiTect SYBR Green RT-PCR kit (Qiagen, Valencia, CA, USA). The relative expression was calculated using the Power SYBR™ Green PCR Master Mix (Applied Biosystems, Foster City, CA, USA). Primers used in qRT-PCR are listed in Appendix A. All experiments were repeated at least three times.

### 2.4. Cell Culture and Reagents

The normal ovarian surface epithelium, the HOSEpic cell line (cat#7310), was purchased from ScienCell (Carlsbad, CA, USA). The clear cell line, ES-2 (cat#CRL-1978) and TOV21G (cat#CRL-11730) cell lines, were purchased from ATCC (Manassas, VA, USA). High-grade serous carcinoma (HGSC), Caov3 (cat#30075), OVCAR3 (cat#30161), and SKOV3 (cat#30077) cell lines were purchased from the Korean Cell Line Bank (Seoul, Korea). The HOSEpic cell line was cultured in an ovarian epithelial cell medium (cat#7311, ScienCell) supplied with an ovarian epithelial cell growth supplement (cat#7352, ScienCell) and an antibiotic solution (cat#0503, ScienCell). All cell lines have been authenticated using STR profiling. The ES-2, TOV21G, OVCAR3, and SKOV3 cell lines were cultured in RPMI (cat# 11875093, Thermo Fisher Scientific, Waltham, MA, USA), supplemented with 10% fetal bovine serum (cat# 35015CV, Thermo Fisher Scientific), 1% penicillin, and streptomycin. The GSI, DAPT (N-[N-(3,5-difluorophenacetyl)-L-alanyl]-S-phenylglycine t-butyl ester), was purchased from Sigma-Aldrich (cat# D5942, Burlington, MA, USA). For DAPT treatment, cells were cultured for 48 h at the concentration indicated in the figures.

### 2.5. siRNA Transfection and Western Blot Analysis

Three different small interfering RNAs (siRNAs) targeting NOTCH1 (cat#4851-1, 4851-2, and 4851-3) and NOTCH3 (cat#4854-1, 4854-2, and 4854-3) were purchased from Bioneer (Deajeon, Korea). Non-targeting siRNA was used as a negative control. The RNAi oligonucleotide or RNAi negative control was transfected into the cells using a Lipofectamine RNAiMAX Transfection Reagent (Thermo Fisher Scientific, Waltham, MA, USA) according to the manufacturer’s instructions. Cells were lysed with a cell extraction buffer (Thermo Fisher Scientific) containing protease inhibitors. Protein concentrations were determined using the BCA protein assay (Thermo Fisher Scientific). Protein samples were separated using SDS-PAGE and transferred to polyvinylidene difluoride membranes. The membranes were incubated with the indicated primary antibodies overnight at 4 °C. The following antibodies were used: NOTCH1 (cat#4380, Cell Signaling Technology, Danvers, MA, USA), NOTCH3 (cat#5276, Cell Signaling Technology), and α-tubulin (cat#sc-8035, Santa Cruz Biotechnology, Inc., Dallas, TX, USA). The blots were visualized using the Pierce™ ECL Western Blotting Substrate (Thermo Fisher Scientific, Waltham, MA, USA). The original WB can be found at Appendix A.

### 2.6. Total RNA Sequencing

Total RNA was isolated from OVCAR3 cells using Trizol (Thermo Fisher Scientific). After confirming RNA quality, the TruSeq Stranded mRNA LT Sample Prep Kit (Illumina, San Diego, CA, USA) was used to prepare a cDNA library according to the Sample Preparation Guide (Part #15031048 Rev E) by Macrogen Inc (Seoul, Korea). The 100-nt paired-end sequencing was conducted using NovaSeq 6000 (Illumina, San Diego, CA, USA). To remove any bias in the analysis, low-quality RNA or artifacts, such as adaptor sequences, contaminant DNA, and PCR duplicates were trimmed from raw data. Trimmed data was mapped to the reference genome using HISAT2 (version 2.1.0) and the aligned reads were generated. StringTie (version 2.1.3b) was used to produce reconstructions of genes and estimated expression levels using the reference-based aligned reads information. In all, 60,606 transcripts were classified based on their overlap with the GENCODE annotation (version 38).

### 2.7. Statistical Analysis

Heatmaps and correlation graphs were drawn using R studio (version 3.3.3). Continuous variables were compared using Student’s *t*-tests or Mann–Whitney U tests, accordingly. Categorical variables were compared using either χ^2^ tests or Fisher’s exact tests, accordingly. Pearson’s correlation coefficient was used to examine the relationship between lncRNAs with the genes of interest. Statistical analyses were performed using Microsoft Excel (ver. 2016, Microsoft, Redmond, WA, USA), SPSS version 25 for Windows (IBM Corporation, Armonk, NY, USA) and GraphPad Prism (GraphPad Software, Inc., San Diego, CA, USA). Data are presented as the means ± SD, and all *p*-values are two-sided. *p* < 0.05 was considered significant.

## 3. Results

### 3.1. NOTCH1/3 and Their Target Gene Expression in TCGA HGS-OVCA

To discover potential Notch-related lncRNAs, we used the public data from TCGA HGS-OVCA (*n* = 303) and analyzed it according to the NOTCH1/3 expression status. First, we classified the dataset into a high group with a high expression of both NOTCH1 and NOTCH3 (*n* = 97), and a low group with a low expression of both NOTCH1 and NOTCH3 (*n* = 96). The criteria for high and low expression levels were based on median transcripts per million (TPM)-mapped read values. To validate whether these group classifications reflected the activity of Notch signaling, we generated a Notch target gene set (*n* = 36) based on a literature review and performed GSEA using our homemade gene set [24]. As shown in Appendix A, the high NOTCH1/3 group showed a coordinated enrichment of Notch target genes (*p* < 0.001). Following the validation of our group classification, we analyzed the expression of the entire gene set and generated a heatmap using 100 representative genes showing the greatest difference in expression between the two groups (Appendix A). This heatmap clearly suggests that Notch signaling contributes to the up- or down-regulation of diverse genes. To understand the cell biological changes by Notch signaling, we performed GSEA using KEGG and Hallmark gene sets (Appendix A and Figure 1A). As expected, a gene set related to Notch signaling was coordinately enriched in the high NOTCH1/3 group, as shown by the analyses of both gene sets (Appendix A and Figure 1A). Besides Notch signaling, gene sets related to WNT signaling, epithelial mesenchymal transition (EMT), and Hedgehog signaling were highly enriched in the high NOTCH1/3 group (Figure 1A). The results suggested that Notch signaling could affect invasive features of HGSC by orchestrating EMT-related gene expression; our classification appropriately reflected the activity of Notch-signaling.

### 3.2. Discovery of Notch-Related lncRNAs Using TANRIC Database

Ten of the available TCGA HGS-OVCA cases (*n* = 303) did not have data related to the lncRNA expression status, and the remaining 293 cases were analyzed to match mRNA and lncRNA expression levels. First, we divided the HGS-OVCA cases into four groups: I (*n* = 92), the low expression of both NOTCH-1 and -3; II (*n* = 54), the high expression of NOTCH3 and the low expression of NOTCH1; III (*n* = 54), the high expression of NOTCH1 and the low expression of NOTCH3; IV (*n* = 93), the high expressions of both NOTCH1 and NOTCH3. Next, based on this grouping, we analyzed the expression status of lncRNAs and generated a heatmap using representative lncRNAs (*n* = 1737), showing a differential gene expression pattern among the four groups (Figure 1B). To investigate the biological impact of the discovered lncRNAs in HGS-OVCA, we used FuncPred, a web-based in silico predictor of lncRNA functions. Applying the top 10% highly expressed lncRNAs in Group IV to FuncPred, we found 46 related Hallmark gene sets (Figure 1C). Interestingly, every 13th gene set significantly enriched in the high NOTCH1/3 expression group of our GSEA overlapped with the lncRNA-related gene sets in FuncPred (Figure 1D). This data suggested that Notch-related lncRNA may cooperate with Notch-related mRNA to generate the invasiveness of HGSC; again, our grouping and analytic method appropriately reflected Notch signaling activation. Among the 33 gene sets discovered only in FuncPred, the top-ranked gene sets related to DNA repair and spermatogenesis were not significantly enriched in the high NOTCH1/3 expression group by GSEA using mRNA data from TCGA OVCA. Taken together, these data suggested that Notch-related lncRNAs could be involved in complex biological functions in Notch up-regulated ovarian cancer, along with Notch-related mRNA, and the functional prediction of lncRNAs might be a powerful tool to investigate NOTCH1/3 functions in HGSC.

### 3.3. Validation of Notch-Related lncRNAs by NOTCH1/3 Silencing and Analysis of Clinical Tissue Samples

To validate our FuncPred analysis results, we investigated the effect of NOTCH1/3 silencing using siRNAs. First, we performed a Western blot analysis to check the NOTCH1/3 expression status in various normal ovarian cell lines and ovarian cancer cell lines (Appendix A). Among the cell lines tested, TOV21G and OVCAR3 showed simultaneous NOTCH1 and NOTCH3 expressions. Because TCGA OVCA includes data from patients with HGSC but not from clear cell carcinomas, we selected the HGSC cell line, OVCAR3, for this study. We tested three types of siNOTCH1 and siNOTCH3 (Appendix A, respectively). These efficiently reduced transmembrane NOTCH1 or NOTCH3, as analyzed by Western blotting (Appendix A). The total RNAs extracted from OVCAR3 treated with two kinds of siRNA combinations were then subjected to total RNA sequencing. In all, 550 differentially expressed lncRNAs were identified (Figure 2A). From these, we selected 216 lncRNAs that were downregulated by NOTCH1/3 silencing to investigate the related gene sets using FuncPred. The lncRNA-related gene sets obtained from the FuncPred analysis (Figure 2B) are similar to those from our TANRIC analysis (Figure 1C). We found 43 Notch-related lncRNAs in common between the 1670 up-regulated lncRNAs in high NOTCH1/3 OVCA and the 216 lncRNAs that were down-regulated by NOTCH1/3 silencing (Figure 2C,D). The FuncPred analysis, using these 43 lncRNAs, revealed that genes related to spermatogenesis and DNA repair were consistently top ranked among the related genes (Figure 2E).

To validate our transcriptomic data, we silenced NOTCH1/3 in OVCAR3 cells using different combinations of siRNAs specific for NOTCH1 and NOTCH3 (Appendix A). Using total RNAs from these cells, we detected a decreased NOTCH1/3 expression, along with their well-known target genes, such as the hairy and enhancer-of-split-1 (HES1) and the hairy/enhancer-of-split related to YRPW motif protein 1 (HEY1) (Appendix A). The levels of representative common lncRNAs, such as AP000525.1, ASB16-AS1, PRKCZ-AS, GUSBP11, and AL355488.1 were also decreased by different combinations of siRNAs (Figure 3A). In addition, genes related to DNA repair and spermatogenesis were also decreased by these siRNAs (Figure 3B,C). Mature Notch receptors are processed and assembled as heterodimeric proteins [33]. The binding of Notch receptors to ligands triggers the dissociation of heterodimeric proteins, which causes the cleavage of the receptors by γ-secretase. These result in the release of the Notch intracellular domain (NICD), which, in turn, translocates to the nucleus and activates the transcription of Notch targets [34,35]. DAPT is a potent γ-secretase inhibitor (GSI) that targets Notch signaling. DAPT diminishes the breakdown product transmembrane/intracellular region which reflects the activity of Notch in a dose-dependent manner (Appendix A). Treating OVCAR3 with DAPT, we observed a decreased expression of NOTCH1/3, along with their target genes HES1 and HEY1 (Appendix A). Representative Notch-related lncRNAs and genes related to DNA repair and spermatogenesis were also down-regulated in OVCAR3 following DAPT treatment (Figure 3D–F).

To validate these results in clinical tissue samples, we extracted RNAs from HGSC (*n* = 79) and divided them into two groups: the high NOTCH1/3 expression group (*n* = 21) and the low NOTCH1/3 expression group (*n* = 19) (Figure 4A). We observed the increased expression of representative Notch-related lncRNAs and genes related to DNA repair and spermatogenesis (Figure 4B,C). On the other hand, the analysis of benign or borderline ovarian tumors (*n* = 16) and clear cell ovarian cancers (*n* = 5) identified low NOTCH1/3 expressions and also presented low expressions of Notch-related lncRNAs and their related gene sets, compared to those in HGSC (*n* = 79) (Figure 4D–G). In summary, all our validation experiments and the FuncPred analysis clearly supported our analysis using TANRIC database, showing that Notch-related lncRNAs and genes were frequently upregulated in HGSC.

### 3.4. Simultaneous Expression of Common lncRNAs and Genes Related to DNA Repair and Spermatogenesis

Because the transcription of lncRNA is simultaneously accompanied by the expression of subsets of genes, we performed correlation analyses between the 43 common lncRNAs and genes related to DNA repair (*n* = 150) and spermatogenesis (*n* = 135) using TANRIC and our total transcriptome data from siRNA experiments. Whereas a large number of genes and lncRNAs showed significant positive correlations, the analysis of our total transcriptome data from the siRNA experiments presented higher correlation coefficients (Figure 5A,B), suggesting a direct relationship in their expression mechanisms. Based on these findings, we sought to reveal the TFs commonly utilized by the 43 lncRNAs and genes related to DNA repair and spermatogenesis, using ENCODE (Figure 5C). Although only 21 out of 43 lncRNAs could be evaluated, we were able to discover common TFs for them, including Early Growth Response 1 (EGR1), CCCTC-binding factor (CTCF), GA-binding protein α (GABPA), and E2F Transcription Factor 4 (E2F4) (Figure 5D). The genes related to DNA repair and spermatogenesis also utilize these TFs (Figure 5E). GSEA also showed that transcriptional targets of EGR1 and E2F4 were highly enriched in the high NOTCH1/3 group in TCGA OVCA and in the siNOTCH1/3 group in our transcriptome data. However, the target gene sets of CTCF and GABPα could not be analyzed as these TFs were not provided by MSigDB (Figure 5F). Taken together, our correlation analyses and the identification of common TFs using ENCODE suggested that Notch-related lncRNAs may share master TFs and work cooperatively with their related gene sets.

## 4. Discussion

Ovarian cancer, especially high-grade serous ovarian carcinoma, is one of the most lethal human malignancies [2,10,24]. Patients with ovarian cancer usually present with advanced disease stages and show frequent recurrences, that are refractory against platinum-based conventional chemotherapy [8]. To overcome these therapeutic limitations, novel targets, such as Notch signaling, have been rigorously investigated [24,36]. Several Notch-targeting drugs, including GSIs, have been tested in phase I and phase II trials [23]. Transcription factors of the Hes and Hes-related (Hey) families are the most well-known targets in the Notch pathway [37]. Cell cycle regulators, cyclin D1 and p21, and NF-κB family members, c-Myc and Deltex, are also included in Notch target genes [38,39,40,41]. As shown by our results from GSEA, the Notch pathway has a cooperative relationship with various signaling pathways, including WNT/β-catenin, TGFβ, and Hedgehog signaling [42,43,44]. This complex crosstalk with other signaling pathways have been reported to highly depend on the cellular context and environment [45]. However, the exact mechanisms of this crosstalk remain to be elucidated.

In addition to its interaction with other signaling pathways, Notch has been reported to induce lncRNA expressions to promote tumor progression in colorectal cancer [26]. The expression of lncRNAs in ovarian cancer has been investigated, suggesting a possible role in tumor progression, estrogen responses, and resistance to conventional chemotherapy [46]. Among the previously reported lncRNAs, some lncRNAs are Notch-related, for example, HOTAIR and MALAT1 [47,48,49,50,51,52,53,54]. However, to the best of our knowledge, this study is the first to investigate Notch-related lncRNAs in ovarian cancer using a genome-wide approach. Using TANRIC database, we identified 1757 differentially expressed lncRNAs based on NOTCH1/3 expressions; 1670 were upregulated, and 87 downregulated. Because data acquired from tumor tissue can be distorted by the complexity of multiple signaling pathways, we tried to confirm our results from TANRIC with siRNA experiments, which allowed the observation of relatively short time changes. After the transfection with NOTCH1/3-target siRNAs, we obtained 550 differentially expressed lncRNAs and selected 216 down-regulated lncRNAs. Finally, a comparative analysis of these downregulated lncRNAs with our data from TANRIC resulted in the identification of 43 common lncRNAs.

An interesting aspect of our functional analysis was that the genes related to specific functions were consistently associated with Notch-related lncRNAs. GSEA using TCGA OVCA revealed the signaling complexity related to Notch signaling. However, our functional analysis using FuncPred suggested more diverse gene sets related to Notch-related lncRNAs. Furthermore, as the number of representative lncRNAs decreased as we proceeded from TANRIC to siRNA experiments, the complexity of these signal changes was reproducible. Even a reduction in the number of representative lncRNAs from 1670 to 43 did not significantly change the results of these functional analyses.

The diverse roles of lncRNA in cell biology have been intensively investigated [55]. However, the regulatory mechanism of lncRNA expression itself has not been well understood. Independent transcriptional regulatory mechanisms might exist, or their expression might be related to DNA methylation [56]. In addition, the regulation of lncRNA expression might occur simultaneously with the process of regulating the coding genes [18,57]. Our correlation analysis supported the simultaneous expression of lncRNAs with coding genes. As we performed the correlation analysis using total transcriptome data from siRNA experiments, more than 50% of genes related to DNA repair and spermatogenesis presented high correlation coefficients, indicating the simultaneous expression of lncRNAs and coding genes. In fact, the FuncPred algorithm for predicting the lncRNA functions is based on the proximity of the lncRNAs and genes in the chromosome. Our analysis using ENCODE also supported this hypothesis. The 21 lncRNAs out of 43 investigated in ENCODE showed putative common TFs, such as EGR1, CTCF, GABPα, and E2F4, which were likewise found in novel gene sets related to DNA repair and spermatogenesis. In addition, our GSEA showed that the transcriptional targets of these common TFs were coordinately enriched in the high NOTCH1/3 and siNOTCH1/3 groups, indicating that the molecular targets in Notch upregulated ovarian cancer might be orchestrated by these master TFs [58,59,60,61].

## 5. Conclusions

In this study, we investigated the Notch-related lncRNAs using an in silico analysis and in vitro experiments, followed by validation using clinical samples in human ovarian cancers. Through this, we identified novel lncRNAs related to Notch signaling and analyzed their close relationship with gene sets involved in functional aspects, such as DNA repair and spermatogenesis, as well as regulatory mechanisms of expression. The results from this study enhance our understanding of the role of lncRNAs in the regulation of Notch signaling and their target genes in ovarian cancer. Although our data did not reveal the regulatory mechanism of master TFs, the precise regulatory mechanisms of Notch-related lncRNAs should be elucidated in the future with further research.

## Figures and Tables

**Figure 1 cancers-14-01557-f001:**
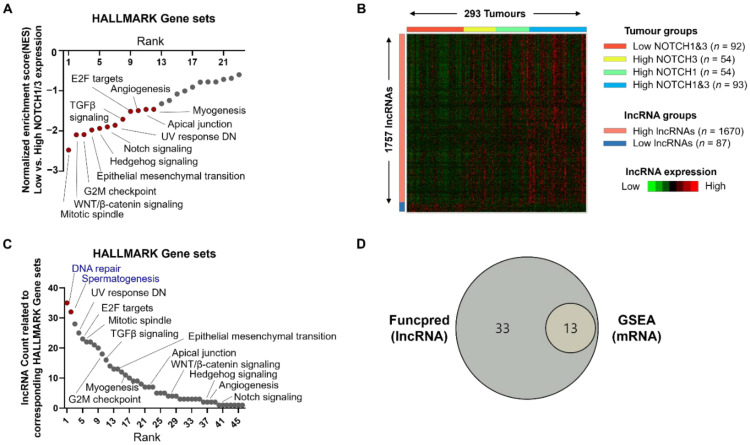
Discovery of Notch-related lncRNAs in HGSC. (**A**) Results of GSEA using Hallmark gene sets in the TCGA HGS-OVCA database (N = 303) to validate our grouping according to NOTCH1/3 expression. “High” indicates a group with high expressions of both NOTCH1 and NOTCH3 (*n* = 97). “Low” indicates a group with low expressions of both NOTCH1 and NOTHC3 (*n* = 96). The classification into high and low was based on median TPM values. High NOTCH1 with low NOTCH3 (*n* = 55) and high NOTCH3 with low NOTCH1 (*n* = 55) were excluded from this analysis. Red coloured dots indicated statistically significant *p*-value (<0.05) and FDR q-value (<0.025). (**B**) Identification of differentially expressed lncRNAs using TANRIC database (*n* = 293). Ten of the TCGA HGS-OVCA cases (*n* = 303) did not show lncRNA expression. Heatmap showing representative lncRNAs (*n* = 1757) according to NOTCH1/3 expression status. Based on the expression status of NOTCH1/3, TCGA HGS-OVCA dataset was divided into four groups: I (*n* = 92), low expression of both NOTCH1 and 3; II (*n* = 54), high expression of NOTCH3 and low expression of NOTCH1; III (*n* = 54), high expression of NOTCH1 and low expression of NOTCH3; and IV (*n* = 93), high expression of both NOTCH1 and 3. (**C**) Potential function of Notch-related lncRNAs using FuncPred. Red coloured dots indicate unique gene sets highly ranked in this analysis but not in GSEA depicted in (**A**). (**D**) Venn diagram presenting the logical relation between FuncPred data with lncRNAs and GSEA data with mRNAs. Abbreviations: NOTCH1/3, Notch receptor 1 and 3; TCGA, The Cancer Genome Atlas; HGS-OVCA, high-grade serous ovarian cancer; lncRNA, long non-coding RNA; GSEA, gene set enrichment analysis; FuncPred, in silico prediction of gene function for protein-coding and lncRNA human genes.

**Figure 2 cancers-14-01557-f002:**
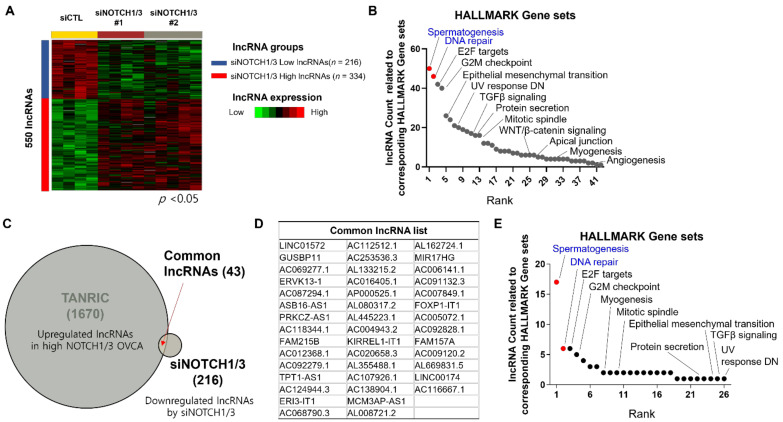
Discovery of Notch-related lncRNAs using transcriptome data from OVCAR3 cells treated with siNOTCH1/3. (**A**) Heatmap showing differentially expressed lncRNAs (*n* = 550) according to siNOTCH1/3 treatment. (**B**) Predicted function of Notch-related lncRNAs based on our siRNA experiments using FuncPred. Red coloured dots indicated unique gene sets highly ranked in this analysis, but not in GSEA, as depicted in Figure 1A. (**C**) Venn diagram presenting the logical relation between upregulated lncRNAs in high NOTCH1/3 group in TCGA OVCA and downregulated lncRNAs in our siRNA experiments. (**D**) List of common lncRNAs, as depicted in (**C**). (**E**) Potential function of the common lncRNAs listed in (**D**). Red coloured dots indicate unique gene sets highly ranked in this analysis, but not in GSEA, as depicted in Figure 1A. Abbreviations: TANRIC, The Atlas of Noncoding RNAs in Cancer.

**Figure 3 cancers-14-01557-f003:**
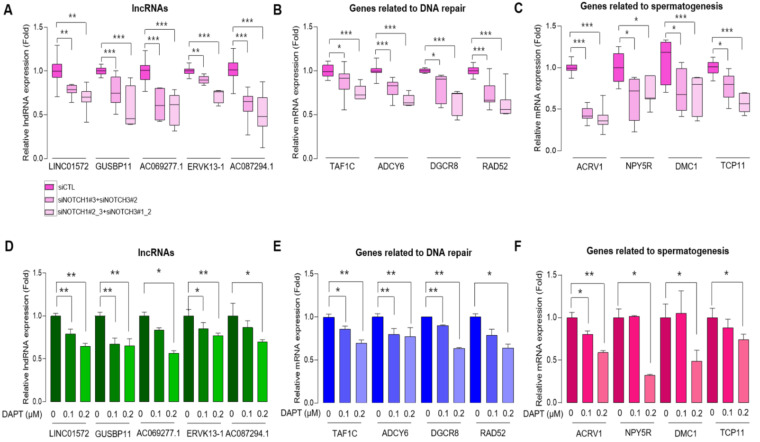
Validation of in silico analysis and transcriptome data in HGSC cell line. Results of qRT-PCR for mRNAs of representative Notch-related lncRNAs (**A**), DNA repair-related targets (**B**), and Spermatogenesis-related targets (**C**) in OVCAR3 cells transfected with control siRNA (siControl) or NOTCH 1/3 siRNAs as indicated for 48 h. Comparisons of the relative expression of representative Notch-related lncRNAs (**D**), genes related to DNA repair (**E**), and spermatogenesis (**F**) in OVCAR3 cells following treatment with DAPT for 48 h. Data is shown as mean ± SD and *p*-values were calculated by two-tailed Mann–Whitney U-test. All experiments were repeated three times, and each experiment was performed in triplicate. * *p* < 0.05, ** *p* < 0.01, *** *p* < 0.001. Abbreviations: CTL, Control; TAF1C, TATA-Box Binding Protein Associated Factor; ADCY6, Adenylate Cyclase Type 6; DGCR8, DiGeorge Syndrome Critical Region Gene 8; RAD52, DNA Repair Protein RAD52 Homolog; ACRV1, Acrosomal Vesicle Protein 1; NPY5R, Neuropeptide Y Receptor Y5; DMC1, DNA Meiotic Recombinase 1; TCP11, T-Complex 11, DAPT: N-[N-(3,5-difluorophenacetyl)-L-alanyl]-S-phenylglycine t-butyl ester.

**Figure 4 cancers-14-01557-f004:**
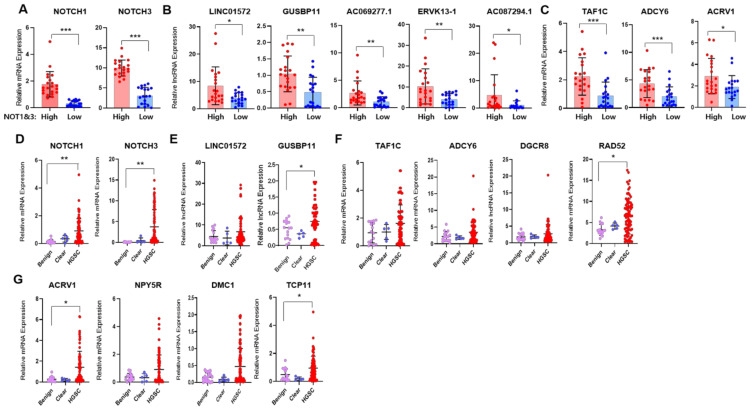
Validation of in silico analysis and transcriptome data in human ovarian tissue samples. (**A**–**C**) Comparisons of the relative expression of NOTCH1/NOTCH3 (**A**), representative Notch-related lncRNAs (**B**), genes related to DNA repair (TAF1C and ADCY6) and spermatogenesis (**C**). (ACRV1) using our own HGSC tissue samples according to NOTCH1/3 expression (high vs. low, *n* = 21 vs. *n* = 19, respectively). (**D**–**G**). Relative expression values of NOTC1/3 (**D**), notch-related lncRNAs (**E**), and representative genes related to DNA repair (**F**), and spermatogenesis (**G**), in benign or borderline ovarian tumors (*n* = 16), clear cell ovarian carcinomas (*n* = 5), and high-grade serous ovarian cancers (HGSC, *n* = 79). Data was shown as mean ± SD and *p*-values were calculated by two-tailed Mann–Whitney U-test. All experiments were repeated three times, and each experiment was performed in triplicate. * *p* < 0.05, ** *p* < 0.01, *** *p* < 0.001. Abbreviations: TAF1C, TATA-Box Binding Protein Associated Factor; ADCY6, Adenylate Cyclase Type 6; DGCR8, DiGeorge Syndrome Critical Region Gene 8; RAD52, DNA Repair Protein RAD52 Homolog; ACRV1, Acrosomal Vesicle Protein 1; NPY5R, Neuropeptide Y Receptor Y5; DMC1, DNA Meiotic Recombinase 1; TCP11, T-Complex 11.

**Figure 5 cancers-14-01557-f005:**
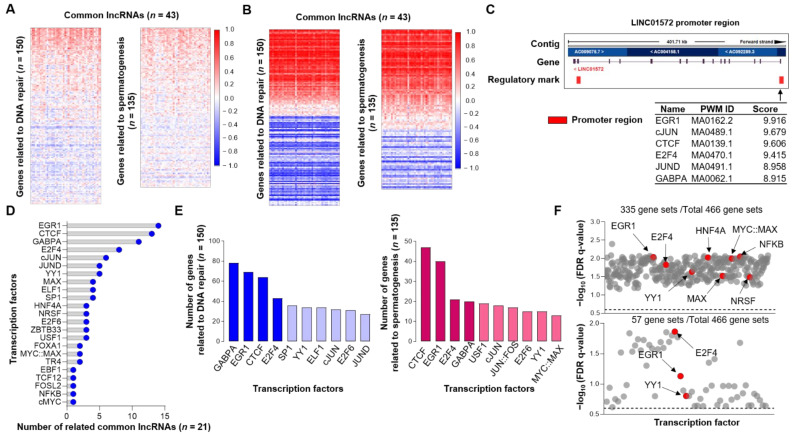
Identification of common transcription factors for Notch-related lncRNAs and genes related to DNA repair and spermatogenesis. (**A**) Correlation analysis of common lncRNA clusters with genes related to DNA repair and spermatogenesis using data from TCGA OVCA and (**B**) our siRNA experiments. (**C**) Representative result of ENCODE predicting putative transcription factors. (**D**) Results from ENCODE using 21 out of 43 common lncRNAs presenting number of lncRNAs associated with putative transcription factors. (**E**) Results from ENCODE using genes related to DNA repair and spermatogenesis presenting number of genes associated with putative transcription factors. (**F**) Transcription factors that are up- or down-regulated according to NOTCH1/3 status. GSEA was performed using legacy subset of transcription target gene sets as described in the Methods section. Upper and lower panels presented the results of GSEA using TCGA OVCA and our transcriptome data, respectively. The legacy subset of transcription target gene sets originally supported 610 gene sets, but we excluded unknown transcription factor and applied gene set size filter (min = 15, max = 500), and finally used 466 gene sets. Each dot represents one transcription target gene set which showed adjusted *p*-value < 0.05 and FDR q-value < 0.25. The dotted line indicates FDR = 0.25. Abbreviations: EGR1, Early Growth Response 1; CCTF, CCCTC-Binding Factor; GABPA, GA Binding Protein Transcription Factor Subunit Alpha; E2F4, E2F Transcription Factor 4; FDR, False discovery rate.

## Data Availability

The data presented in this study are available on request from the corresponding author.

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
