# Peer review of "Long Non-Coding RNA-Based Functional Prediction Reveals Novel Targets in Notch-Upregulated Ovarian Cancer"

_cancers, 2022, doi:10.3390/cancers14061557_

Round 1

Reviewer 1 Report

Comments on Jeong et al:

The aim of this manuscript is to perform an in silico analysis, using The Cancer Genome Atlas data, in order to individuate novel Notch-related lncRNAs in ovarian cancer (OVCA). The main question addressed by this research is to deeply analyze the functional linkage between Notch-related lncRNAs and Notch related transcripts, in the context of ovarian cancer.

As regards the main topic, it is interesting and certainly of great scientific and clinical impact: in fact, this manuscript touches a significant area, fousing on the pivotal role played by LncRNSa in the pathogenesis and progression of ovarian cancer. As regards the originality and strenghts of this manuscript, this is a significant contribute to the ongoing research on this topic, as it extends the research field on the oncogenic action of LncRNAs in ovarian cancer. Overall, the contents are rich, and the authors also give their deep insight for some works.

At the same time, this manuscript show rich content, providing a deep insight for some works: I found it to be well-written and accessible, providing sufficient information for the non-expert while also achieving a balance of detail for those with more expertise in the field. This is the additional point, which makes this manuscript original, in comparison to published literature.

As regards the section of methods, there is a specific and detalied explanation for the majority of methods used in this study: this is particularly significant, since the manuscript relies on a multitude of methodological and statistical analysis, to derive its conclusions. The methodology applied is overall correct, the results are reliable and adequately discussed.

The conclusion of this manuscript is perfectly in line with the main purpouse of the paper: the authors have designed and conducted the study properly and they also discuss the limitation of this work, as regards the regulatory mechanism of master TFs, key point for future research. As regards the conclusions, they are well written and present an adequate balance between the description of previous findings and the results presented by the authors.

The minor point of this paper is associated to the reference: as mentioned before, the manuscript will benefit from highlighting that HGSC is considered, due to structural and ultrastructural evidence, the most aggressive subtype of ovarian cancer, with a debated origin and poor prognosis. In this context, the authors should introduce this aspect, in order to provide to the readers both a morphological and molecular profile of this subtype of ovarian cancer (see, for reference: Giusti, I., Bianchi, S., Nottola, S. A., Macchiarelli, G., Dolo, V. (2019). CLINICAL ELECTRON MICROSCOPY IN THE STUDY OF HUMAN OVARIAN TISSUES. EuroMediterranean Biomedical Journal14). At the same time, it will be useful to remark how several lncRNAs, among them HOTAIR (Hox transcript antisense RNA), enhances the invasion and migration of ovarian cancer cells, with significant and universal functions in the tumorigenesis, as suggested by several and recent studies (see, for reference: Dai ZY, Jin SM, Luo HQ, Leng HL, Fang JD. LncRNA HOTAIR regulates anoikis-resistance capacity and spheroid formation of ovarian cancer cells by recruiting EZH2 and influencing H3K27 methylation. Neoplasma. 2021 May;68(3):509-518. doi: 10.4149/neo_2021_201112N1212. Epub 2021 Jan 28. PMID: 33502891).

Finally, this manuscript also presents a basic structure, properly divided and characterized by organic and detailed figures and tables. This manuscript looks like very informative since there are few evidence on this topic and is sound in highlighting the role of LncRNAs in the context of ovarian cancer.

Author Response

The aim of this manuscript is to perform an in silico analysis, using The Cancer Genome Atlas data, in order to individuate novel Notch-related lncRNAs in ovarian cancer (OVCA). The main question addressed by this research is to deeply analyze the functional linkage between Notch-related lncRNAs and Notch related transcripts, in the context of ovarian cancer.

As regards the main topic, it is interesting and certainly of great scientific and clinical impact: in fact, this manuscript touches a significant area, fousing on the pivotal role played by LncRNSa in the pathogenesis and progression of ovarian cancer. As regards the originality and strenghts of this manuscript, this is a significant contribute to the ongoing research on this topic, as it extends the research field on the oncogenic action of LncRNAs in ovarian cancer. Overall, the contents are rich, and the authors also give their deep insight for some works.

At the same time, this manuscript show rich content, providing a deep insight for some works: I found it to be well-written and accessible, providing sufficient information for the non-expert while also achieving a balance of detail for those with more expertise in the field. This is the additional point, which makes this manuscript original, in comparison to published literature.

As regards the section of methods, there is a specific and detalied explanation for the majority of methods used in this study: this is particularly significant, since the manuscript relies on a multitude of methodological and statistical analysis, to derive its conclusions. The methodology applied is overall correct, the results are reliable and adequately discussed.

The conclusion of this manuscript is perfectly in line with the main purpouse of the paper: the authors have designed and conducted the study properly and they also discuss the limitation of this work, as regards the regulatory mechanism of master TFs, key point for future research. As regards the conclusions, they are well written and present an adequate balance between the description of previous findings and the results presented by the authors.

R1_Q1> The minor point of this paper is associated to the reference: as mentioned before, the manuscript will benefit from highlighting that HGSC is considered, due to structural and ultrastructural evidence, the most aggressive subtype of ovarian cancer, with a debated origin and poor prognosis. In this context, the authors should introduce this aspect, in order to provide to the readers both a morphological and molecular profile of this subtype of ovarian cancer (see, for reference: Giusti, I., Bianchi, S., Nottola, S. A., Macchiarelli, G., Dolo, V. (2019). CLINICAL ELECTRON MICROSCOPY IN THE STUDY OF HUMAN OVARIAN TISSUES. EuroMediterranean Biomedical Journal, 14). At the same time, it will be useful to remark how several lncRNAs, among them HOTAIR (Hox transcript antisense RNA), enhances the invasion and migration of ovarian cancer cells, with significant and universal functions in the tumorigenesis, as suggested by several and recent studies (see, for reference: Dai ZY, Jin SM, Luo HQ, Leng HL, Fang JD. LncRNA HOTAIR regulates anoikis-resistance capacity and spheroid formation of ovarian cancer cells by recruiting EZH2 and influencing H3K27 methylation. Neoplasma. 2021 May;68(3):509-518. doi: 10.4149/neo_2021_201112N1212. Epub 2021 Jan 28. PMID: 33502891).

Finally, this manuscript also presents a basic structure, properly divided and characterized by organic and detailed figures and tables. This manuscript looks like very informative since there are few evidence on this topic and is sound in highlighting the role of LncRNAs in the context of ovarian cancer.

R1_A1> We appreciate the reviewer’s comments. As the reviewer suggested, we have added relevant references to the introduction section of the manuscript (line 48-50, line 61-63). We also believe that the papers suggested by the reviewers are very helpful in understanding the purpose and direction of our paper. Once again, thank the reviewer for your interest in and careful review of our paper.

Reviewer 2 Report

Authors have identified NOTCH-dependent lncRNAs in ovarian cancer. The results are clearly and well represented. 

  1. What is the main question addressed by the research?
    Ovarian cancer is deadly diseases partly due to lack of diagnostic markers, in this regard authors have performed in silico coupled with molecular biology analyses to identify lncRNAs implicated in NOTCH-signaling, which could be therapeutic targets.
    2. Do you consider the topic original or relevant in the field, and if so, why?
    The manuscript is vital to the field due to its nature of investigation focused on identifying new markers in ovarian cancer.
    3. What does it add to the subject area compared with other published material?
    Treating ovarian cancer is key to combating related mortality among women, and authors have identified key markers related NOTCH-signaling, which could be essential in designing therapeutic strategy.
    4. What specific improvements could the authors consider regarding the methodology?
    Post transcription assay analyses could be more elaborate.
    5. Are the conclusions consistent with the evidence and arguments presented and do they address the main question posed?
    Conclusions are consistent with the results, as authors themselves pointed out that additional investigation is required to validate these predictions.
    6. Are the references appropriate? References are appropriate!
    7. Please include any additional comments on the tables and figures. Tables and figures are legible and easy to follow.

Author Response

Authors have identified NOTCH-dependent lncRNAs in ovarian cancer. The results are clearly and well represented.

1. What is the main question addressed by the research?

Ovarian cancer is deadly diseases partly due to lack of diagnostic markers, in this regard authors have performed in silico coupled with molecular biology analyses to identify lncRNAs implicated in NOTCH-signaling, which could be therapeutic targets.

2. Do you consider the topic original or relevant in the field, and if so, why?

The manuscript is vital to the field due to its nature of investigation focused on identifying new markers in ovarian cancer.

3. What does it add to the subject area compared with other published material?

Treating ovarian cancer is key to combating related mortality among women, and authors have identified key markers related NOTCH-signaling, which could be essential in designing therapeutic strategy.

4. What specific improvements could the authors consider regarding the methodology?

Post transcription assay analyses could be more elaborate.

5. Are the conclusions consistent with the evidence and arguments presented and do they address the main question posed?

Conclusions are consistent with the results, as authors themselves pointed out that additional investigation is required to validate these predictions.

6. Are the references appropriate? References are appropriate!

7. Please include any additional comments on the tables and figures. Tables and figures are legible and easy to follow.

R2_A1> We thank the reviewer for the positive response. We especially thank for clear understanding the purpose of this paper and the importance of Notch-associated lncRNA in ovarian cancer.

Reviewer 3 Report

This is a very nice paper describing novel lncRNA targets of NOTCH signaling in ovarian cancer. The flow of experiments is very logical and easy to follow: it begins with research of TCGA data, followed by silencing of NOTCH1 and 3 and subsequent RNAseq analysis to identify NOTCH targets. TCGA data and RNAseq data is then compared, several mRNA and lncRNA targets selected, and validated by qPCR. Following all this, the qPCR analysis was also done on a 100 clinical OC samples.

The only minor comment I have is that the SK-OV-3 cell line is actually ovarian serous cystadenocarcinoma, not HGSC (the other two are HGSC as specified, and the experiments were done on the HGSC line, so it's OK).

Author Response

This is a very nice paper describing novel lncRNA targets of NOTCH signaling in ovarian cancer. The flow of experiments is very logical and easy to follow: it begins with research of TCGA data, followed by silencing of NOTCH1 and 3 and subsequent RNAseq analysis to identify NOTCH targets. TCGA data and RNAseq data is then compared, several mRNA and lncRNA targets selected, and validated by qPCR. Following all this, the qPCR analysis was also done on a 100 clinical OC samples.

R3_Q1> The only minor comment I have is that the SK-OV-3 cell line is actually ovarian serous cystadenocarcinoma, not HGSC (the other two are HGSC as specified, and the experiments were done on the HGSC line, so it's OK).

R3_A1> We corrected it as the reviewer pointed out (Supplemental Figure S3A). We thank the reviewer for the positive response.